# Concerted IL-25R and IL-4Rα signaling drive innate type 2 effector immunity for optimal helminth expulsion

Katherine A Smith[1,2]*, Stephan Löser[3], Fumi Varyani[4], Yvonne Harcus[4], Henry J McSorley[4], Andrew NJ McKenzie[5], Rick M Maizels[3]*

[1]Cardiff Institute of Infection and Immunity, Cardiff University, Cardiff, United Kingdom; [2]Institute of Infectious Disease and Molecular Medicine, University of Cape Town, Cape Town, South Africa; [3]Wellcome Centre for Molecular Parasitology, Institute of Infection, Immunity and Inflammation, University of Glasgow, Glasgow, United Kingdom; [4]MRC Centre for Inflammation Research, University of Edinburgh, Edinburgh, United Kingdom; [5]MRC Laboratory of Molecular Biology, Cambridge, United Kingdom

**Abstract** Interleukin 25 (IL-25) is a major 'alarmin' cytokine, capable of initiating and amplifying the type immune response to helminth parasites. However, its role in the later effector phase of clearing chronic infection remains unclear. The helminth *Heligmosomoides polygyrus* establishes long-term infections in susceptible C57BL/6 mice, but is slowly expelled in BALB/c mice from day 14 onwards. We noted that IL-25R (*Il17rb*)-deficient BALB/c mice were unable to expel parasites despite type 2 immune activation comparable to the wild-type. We then established that in C57BL/6 mice, IL-25 adminstered late in infection (days 14–17) drove immunity. Moreover, when IL-25 and IL-4 were delivered to *Rag1*-deficient mice, the combination resulted in near complete expulsion of the parasite, even following administration of an anti-CD90 antibody to deplete innate lymphoid cells (ILCs). Hence, effective anti-helminth immunity during chronic infection requires an innate effector cell population that is synergistically activated by the combination of IL-4Rα and IL-25R signaling.
DOI: https://doi.org/10.7554/eLife.38269.001

*For correspondence:
SmithK28@Cardiff.ac.uk (KAS);
rick.maizels@glasgow.ac.uk (RMM)

**Competing interests:** The authors declare that no competing interests exist.

## Introduction

Type 2 immunity is generated by the immune system in response to a range of environmental challenges from helminth worm parasites, ectoparasites and allergens (*Hammad and Lambrecht, 2015*; *Harris and Loke, 2017*). In the absence of any vaccines, gastrointestinal helminths continue to infect many hundreds of millions of people in less-affluent countries (*Pullan et al., 2014*). Type 2 immunity to parasites is critically dependent upon pathways driven through and co-ordinated by IL-4Rα signaling (*Urban et al., 1998*; *Voehringer et al., 2006*). IL-4Rα activation can result in a multitude of immune outcomes (*Allen and Maizels, 2011*; *Patel et al., 2009*), however, few other components of immunity have been defined as either necessary or sufficient for worm expulsion.

The epithelial-derived 'alarmin' cytokines, including IL-25, IL-33 and TSLP elicit and amplify type 2 immunity through the action of group two innate lymphoid cells (ILC2s) (*Koyasu and Moro, 2012*; *McKenzie et al., 2014*; *Ziegler and Artis, 2010*). There are however important distinctions in both the modes of action of these cytokines, and in the cells, bearing receptors for each mediator. Thus, ligation of the heterodimeric TSLP receptor stimulates dendritic cells to favor a Th2 cell response, and activates innate type 2 populations (*Ito et al., 2005*; *Siracusa et al., 2011*; *Soumelis et al., 2002*; *Ziegler et al., 2013*); IL-25 receptors (heterodimers of IL-17RA and IL-17RB chains) are

expressed on a number of tissues and cell types and increase IL-4 expression by Th2 cells (*Angkasekwinai et al., 2007*); and receptors for IL-33 (an Interleukin-1-like cytokine that signals via the IL-1 receptor-related protein ST2) are broadly expressed in many tissues and induce type 2 cytokine expression by ILC2s and T cells (*Cayrol and Girard, 2018*; *Schmitz et al., 2005*).

Although IL-25 (originally termed IL-17E) shares structural similarity with other IL-17 family members it acts in an opposing manner to IL-17A, by inducing a suite of Type 2 responses (*Kim et al., 2002*; *Pan et al., 2001*) and preventing IL-17A-dependent autoimmune disease (*Kleinschek et al., 2007*). IL-25 is highly expressed within the tuft cells of the gastrointestinal tract (*Gerbe et al., 2016*; *von Moltke et al., 2016*), but is also produced by a range of core immune populations, including T cells and granulocytes. IL-25 plays an important role in inducing allergic inflammation (*Barlow et al., 2012*; *Petersen et al., 2014*) and in driving tissue remodeling and fibrosis (*Gregory et al., 2013*; *Hams et al., 2014*).

A number of laboratories have reported that IL-25 is induced by infection with, and required for, immunity to, the nematode *Nippostrongylus brasiliensis*, a rat parasite that briefly infects immune-competent mice (*Fallon et al., 2006*; *Fort et al., 2001*; *Hurst et al., 2002*; *Zhao et al., 2010*), as well as the nematodes *Heligmosomoides polygyrus* (*Pei et al., 2016*), *Trichuris muris* (*Owyang et al., 2006*) and *Trichinella spiralis* (*Angkasekwinai et al., 2013*). IL-25 may be more important in activation of innate than adaptive immunity, as IL-25-deficient mice mounted normal Th2 responses following *N. brasiliensis* infection (*Mearns et al., 2014*), while ILCs grown in vitro in the presence of IL-25 and IL-33 were able to stimulate expulsion of this parasite upon adoptive transfer to susceptible IL-25R-deficient (*Il17rb$^{-/-}$*) mice (*Neill et al., 2010*).

Due to the rapid nature of expulsion of *N. brasiliensis*, it is difficult to separate the role of IL-25 in the initial drive towards Th2 responsiveness and any subsequent activation of effector mechanisms. We therefore studied chronic infection with the nematode *H. polygyrus*, in which induction over the first 7–10 days can be analysed separately from the subsequent expulsion phase: in so doing, we report that IL-25 is in fact redundant for Th2 initiation, but acts during the later phase of effector expulsion against this helminth parasite. Such a downstream role is consistent with recent reports demonstrating a role for IL-25 in protective immunity to a secondary infection with *H. polygyrus*, following drug clearance of a primary infection (*Pei et al., 2016*), or vaccination with parasite secreted antigens (*Hewitson et al., 2015*). More broadly, a critical role for IL-25 in a late-stage immune response is also evident in airway inflammation models, in which it plays a central role in stimulating inflammatory myeloid cells and promoting airway remodeling (*Gregory et al., 2013*; *Petersen et al., 2012*), or in driving allergic reactions to airway challenge in previously sensitized mice (*Ballantyne et al., 2007*; *Cayrol and Girard, 2018*).

## Results

### IL-25R is redundant for generation of adaptive type-2 responses following chronic helminth infection

To investigate the role of IL-25R in generation of innate and adaptive type-2 responses following chronic parasite infection, we used mice deficient in the IL-25-specific receptor subunit *Il17rb* (*Neill et al., 2010*). We evaluated the outcome of *H. polygyrus* infection in mice of the BALB/c genetic background, which are partially resistant and able to expel most adult worms by day 28 following infection (*Filbey et al., 2014*; *Reynolds et al., 2014*). At day 14 post-infection, shortly after adult worms have matured, *Il17rb$^{-/-}$* mice exhibited significantly increased egg production (*Figure 1A*) but equivalent adult worm burdens (*Figure 1B*) to the IL-25-sufficient wild-type BALB/c. By day 28, the *Il17rb$^{-/-}$* genotype was unable to reduce adult worm or egg burdens (*Figure 1C,D*).

A number of cell types have been shown to express IL-25R and contribute to type-2 inflammation, including ILCs and multi-potent progenitor type-2 cells (*Huang et al., 2015*; *Saenz et al., 2013*), myeloid cells (*Dolgachev et al., 2009*; *Petersen et al., 2012*), NKT cells (*Stock et al., 2009*; *Terashima et al., 2008*) and eosinophils (*Kim et al., 2002*). We first took the approach of analysing individual cell types within BALB/c and *Il17rb$^{-/-}$* mice and found equivalent increases in total and IL-13-expressing ILCs in the mesenteric lymph node (MLN, *Figure 1E,F*) in both strains at day 14 post-infection, and similar increases in the number of Siglec-F$^-$CD11b$^+$Ly6C/G$^+$ myeloid cells in the peritoneal lavage (PL) (*Figure 1G*). However, we noted significantly reduced proportions of Siglec-F$^+$

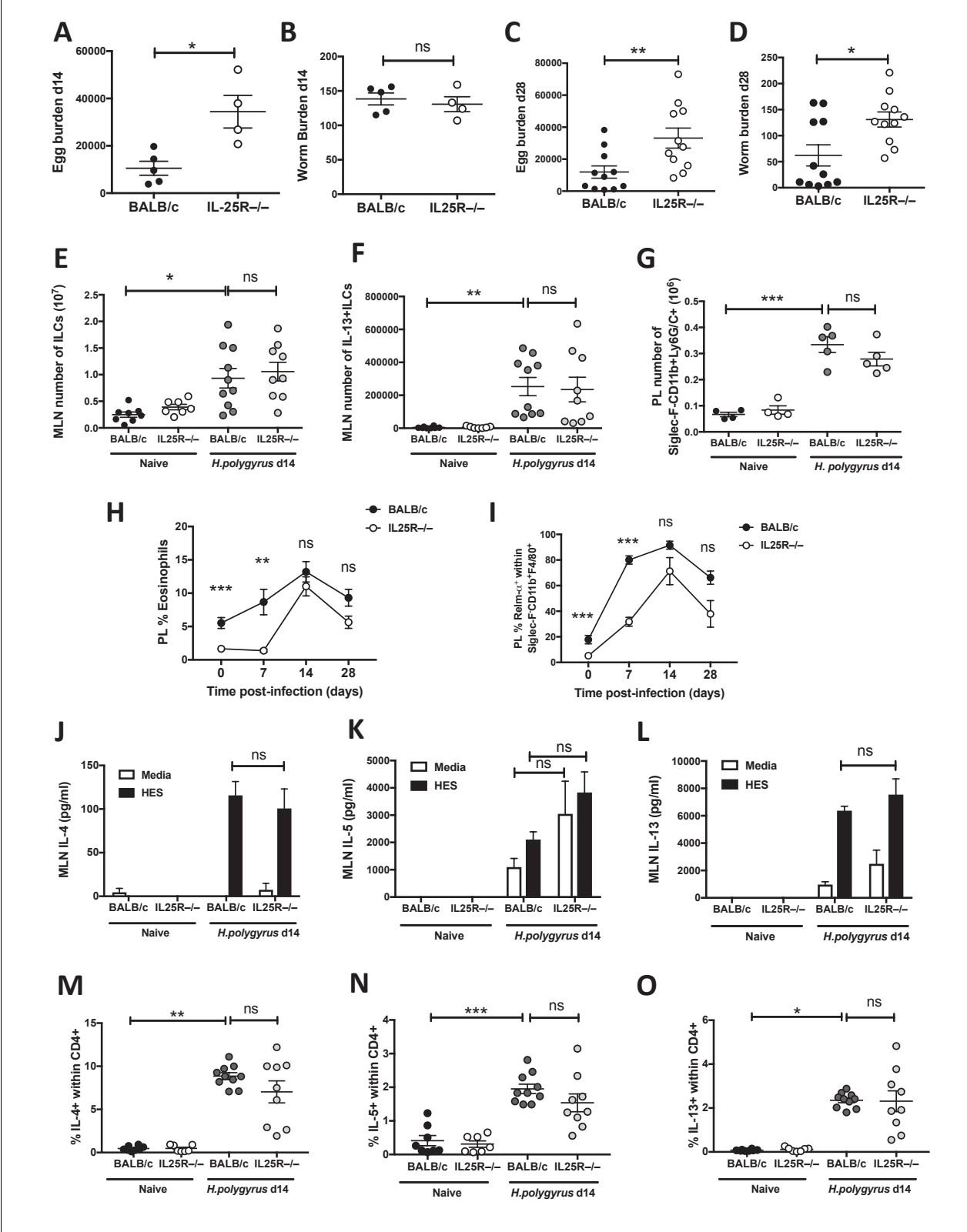

**Figure 1.** IL-25R signaling is required for expulsion of *H. polygyrus* from BALB/c mice. *H. polygyrus*-infected BALB/c or *Il17rb*-deficient (IL25R[−/−]) BALB/c mice were analyzed at day 14 post-infection for fecal egg counts (**A**) and intestinal adult worm burden (**B**) or day 28 post-infection for fecal egg counts (**C**) and intestinal adult worm burden (**D**). MLN cells underwent intracellular cytokine staining (ICCS) to compare the number of Lin[−]ICOS[+]innate lymphoid cells (ILCs) (**E**) and IL-13[+] ILC2s (**F**) by flow cytometry in the different naïve and day 14 infected genotypes. Peritoneal lavage cells (PL) were
*Figure 1 continued on next page*

*Figure 1 continued*

stained with Siglec-F, CD11b and Ly6G/C to compare the number of Siglec-F⁻CD11b⁺Ly6G/C⁺ monocytes (G) in the different naïve and day 14 infected genotypes. Percentages of eosinophils (H) and RELMα⁺ alternatively-activated macrophages (I) were also determined over a 4 week infection timecourse. ELISA of supernatants from MLN cells incubated with media or 1 μg HES for 72 hr was performed to compare IL-4 (J), IL-5 (K) and IL-13 (L) production in the different naïve and day 14 infected genotypes. ICCS of MLN allowed a comparison of the proportion of IL-4⁺, IL-5⁺ and IL-13⁺ CD4⁺ T cells by flow cytometry in the different naïve and day 14 infected genotypes (M–O). Results shown are one representative of three experiments with n ≥ 4 mice/group (A–D), pooled data from two experiments with n ≥ 4 mice/group (C–F, M–O) or one representative of two experiments with n ≥ 4 mice/group (G–L). Data were analysed by unpaired *t* test or one way ANOVA, where *=p≤0.05,**=p≤0.01,***=p≤0.001 and ns = not significant. Error bars represent Standard Error of the Mean.

DOI: https://doi.org/10.7554/eLife.38269.002

eosinophils and resistin-like molecule (RELM)-α⁺ macrophages in the PL of *Il17rb⁻ᐟ⁻* mice at steady-state and during the acute phase of infection (day 7). From day 14 when parasites are in the intestinal lumen and infection enters chronicity, proportions of both cell types increased closer to levels found in wild-type mice (*Figure 1H,I*). We also evaluated typical type 2 cytokine production (IL-4, IL-5 and IL-13) in response to restimulation with parasite antigen (HES); this was found to be equivalent in *H.polygyrus*-infected BALB/c and *Il17rb⁻ᐟ⁻* mice (*Figure 1J–L*). Intracellular cytokine staining of MLN CD4⁺ T cell populations showed no significant difference in expression of IL-4 (*Figure 1M*), IL-5 (*Figure 1N*) or IL-13 (*Figure 1O*) between *H.polygyrus*-infected BALB/c and *Il17rb⁻ᐟ⁻* mice.

## Effective clearance of adult worms requires IL-25R within the hematopoietic immune compartment

During inflammation, expression of IL-25R has also been reported on antigen presenting cells (*Gratchev et al., 2004*), memory T cells (*Wang et al., 2007*), eosinophils (*Tang et al., 2014*) and human vascular endothelial cells (*Wang et al., 2012*) as well as human fibroblasts (*Gregory et al., 2013*). In addition, intestinal smooth muscle hypercontractility is compromised in IL-25-deficient helminth-infected mice (*Pei et al., 2016*). Thus, non-hematopoietic or hematopoietic cell expression of IL-25R might contribute to parasite expulsion in BALB/c mice. To assess this possibility, bone marrow chimeras were generated, infected with *H. polygyrus* and analysed 28 days later for egg and worm burden. Control chimeras reflected the phenotypes of intact mice as *Il17rb⁻ᐟ⁻* reconstituted with *Il17rb⁻ᐟ⁻* bone-marrow had significantly higher worm burdens than BALB/c reconstituted with BALB/

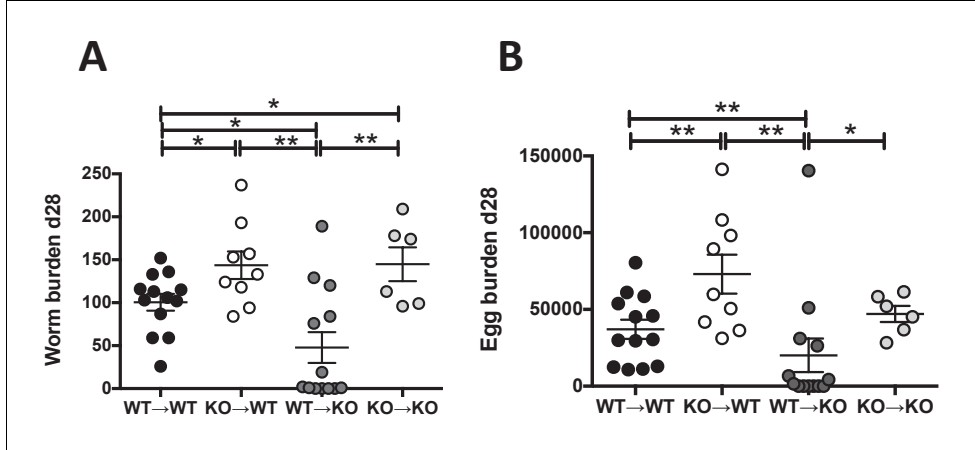

**Figure 2.** IL-25R signaling is required within the hematopoietic compartment for efficient expulsion. Bone marrow chimeras generated from BALB/c (WT) or *Il17rb*-deficient (KO) donor and BALB/c (WT) or *Il17rb*-deficient (KO) recipient mice were infected with *H. polygyrus* and intestinal adult worm burden (A) and fecal egg burden (B) performed at day 28 post-infection. Results shown are pooled data from two experiments performed with n ≥ 3 mice/group, and data from all individual mice are presented. Data were analysed by unpaired *t* test, where *=p≤0.05,**=p≤0.01,***=p≤0.001 and ns = not significant. Error bars represent Standard Error of the Mean.

DOI: https://doi.org/10.7554/eLife.38269.003

c bone-marrow (*Figure 2A*). Efficient adult worm expulsion and decreased egg burden were evident in mice lacking IL-25R on the non-hematopoietic compartment (*Il17rb*$^{-/-}$ reconstituted with BALB/c bone-marrow), however, delayed worm expulsion and increased egg burden occurred in mice lacking IL-25R within the hematopoietic immune compartment (*Figure 2A,B*).

## Effective clearance of adult worms in immune-deficient mice requires IL-25R and IL-4Rα signaling through the innate immune compartment

To test whether stimulation of IL-25R within the innate immune compartment mediates adult worm expulsion and whether this is enhanced following IL-4Rα signaling, immune-deficient *Rag1*$^{-/-}$ mice were infected with *H. polygyrus* and injected with recombinant IL-25 late in infection (d14-17) and/or a complex of rIL-4:anti-IL-4 (IL-4C) on days 13, 16 and 19 post-infection (*Figure 3A*). IL-4C exerted significant but modest reductions in egg counts (44%) and adult worm burden (34%) in *Rag1*$^{-/-}$ mice but did not completely expel adult worms or eliminate egg production (*Figure 3B,C*). Administration of IL-25 alone to *Rag1*$^{-/-}$ mice elicited a downward trend in adult worm numbers, which did not attain statistical significance in two experimental repeats. However when both cytokines were combined, egg production was reduced by 95% and the adult worm burden also greatly reduced (by 87%).

ILC2s may have a role in promoting acquired type-2 immune responses by activation of CD4$^+$ T cell responses through expression of OX40L, MHC class II and PD-L1 (*Drake et al., 2014*; *Mirchandani et al., 2014*; *Oliphant et al., 2014*; *Schwartz et al., 2017*) or by promoting dendritic cell migration to draining lymph nodes following IL-13 production (*Halim et al., 2014*). Sustained activation of ILCs drives immunity to *N. brasiliensis* infection (*Bouchery et al., 2015*), however in *H. polygyrus* infection the transfer of activated ILC2s had only a limited effect on worm establishment (*Pelly et al., 2017*). In addition, IL-25 is able to induce type 2 inflammation and goblet cell hyperplasia in the small intestine, independently of ILCs (*Saenz et al., 2013*). To test whether ILCs contribute to efficient worm expulsion in immune-compromised mice following co-administration of IL-25 and IL-4C, we treated *H. polygyrus* infected *Rag1*$^{-/-}$ mice with both cytokines and isotype or anti-CD90.2 (Thy1.2) antibodies. Administration of anti-CD90.2 antibody significantly reduced the number of CD45$^+$Lin$^-$ ILCs (*Figure 3D,E*) as well as the number of CD45$^+$Lin$^-$ST2$^-$ and CD45$^+$Lin$^-$ST2$^+$ ILCs (*Figure 3F,G*) in the PL of *H. polygyrus*-infected *Rag1*$^{-/-}$ mice treated with IL-25 and IL-4C. However, immunity was fully intact in recipients of this antibody, with significantly decreased parasite egg and worm burden in ILC-depleted mice (*Figure 3H,I*).

## Stimulation of IL-25R induces adult worm clearance late, but not early, in infection

The late manifestation of the *Il17rb*$^{-/-}$ phenotype in *H. polygyrus* infection could reflect a requirement either for initial IL-25 to generate an effector response that is only active after day 14, or for a later IL-25R-driven pathway that is invoked once the Th2 response is generated. To evaluate the relative importance of IL-25R engagement during different phases of infection, we made use of the more susceptible C57BL/6 strain and delivered exogenous rIL-25 early (day 1–4) and late (day 14–17) post-infection (*Figure 4A*). Egg counts were equivalent at day 14 post-infection before delivery of IL-25 late (*Figure 4B*) and were significantly reduced in recipients of IL-25 early (66%; p=0.0013) or late (64%; p=0.0071) compared to PBS controls by day 28 post-infection (*Figure 4C*). Although early IL-25 administration also induced a modest reduction in adult worm burdens, this did not reach significance in two repeated experiments. In contrast, adult worm burden was significantly reduced in recipients of IL-25 late (66%; p=0.0016) at day 28 post-infection (*Figure 4D*). Both CD4$^+$ T cell IL-4 and IL-13 responses (*Figure 4E–F*), and the total number of ILCs (*Figure 4G*) were not significantly altered by late IL-25 injection. The number of IL-13$^+$ILCs (ILC2s) was not significantly increased in the MLN at day 18 *hr. polygyrus* infection but was significantly increased in infected mice following the administration of IL-25 (*Figure 4H*), however, these remained a small proportion of the total ILCs within the MLN

## Monocytes and eosinophils require IL-25R and IL-4Rα signaling for a maximal type-2 response

Monocytes are important regulators of wound repair following Th2 inflammation and are activated to proliferate within the tissue site in response to IL-4Rα signaling, as confirmed in *H. polygyrus*-

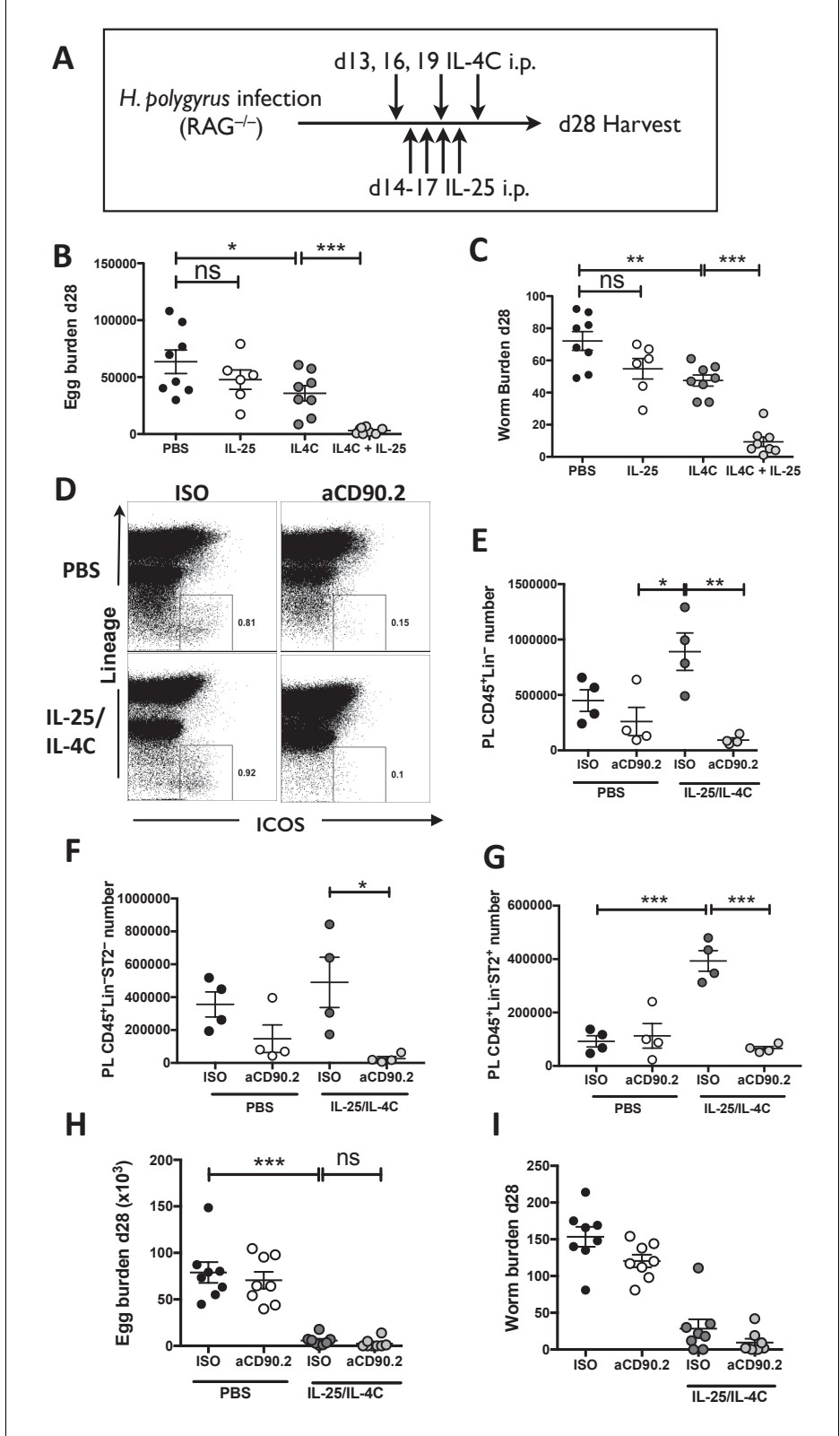

**Figure 3.** IL-25R signaling synergises with IL-4Rα within the innate immune compartment to facilitate efficient worm expulsion. *H. polygyrus*-infected *Rag1*[-/-] mice (RAG[-/-]) were given 0.4 μg recombinant IL-25 i.p. days 14–17 (late) post-infection with or without a complex of 5 μg rIL-4 and 25 μg anti-IL-4 (IL-4C) on days 13, 16 and 19 post-infection, according the schedule shown in (**A**). Mice were analyzed at 28 days post-infection for fecal egg burden

*Figure 3 continued on next page*

*Figure 3 continued*

(B) and intestinal adult worm burden (C). *H. polygyrus*-infected *Rag1⁻/⁻* mice were given IL-25 and IL-4C according to the same schedule, as well as 200 μg of anti-CD90.2/Th1.2 antibody or rat IgG2b control (days 12, 15, 18 and 21). The peritoneal lavage was analyzed at 28 days post-infection for CD45⁺lin⁻ (CD3, CD5, CD8α, CD11c, CD19, DX5, F4/80, GR-1, TCRβ, CD11b), ICOS and ST2 staining by flow cytometry as shown (D) and the number of CD45⁺lin⁻ (E), CD45⁺lin⁻ST2⁻ (F) and CD45⁺lin⁻ST2⁺ (G) ILCs was determined. Mice were analysed at 28 days post-infection for fecal egg burden (H) and intestinal adult worm burden (I). Results shown are one representative of two experiments with n = 4 mice/group (D–G), or pooled data from two experiments with n ≥ 3 mice/group (B,C, H,I). Data were analysed by unpaired *t* test, where *=p≤0.05,**=p≤0.01,***=p≤0.001 and ns = not significant. Error bars represent Standard Error of the Mean.

DOI: https://doi.org/10.7554/eLife.38269.004

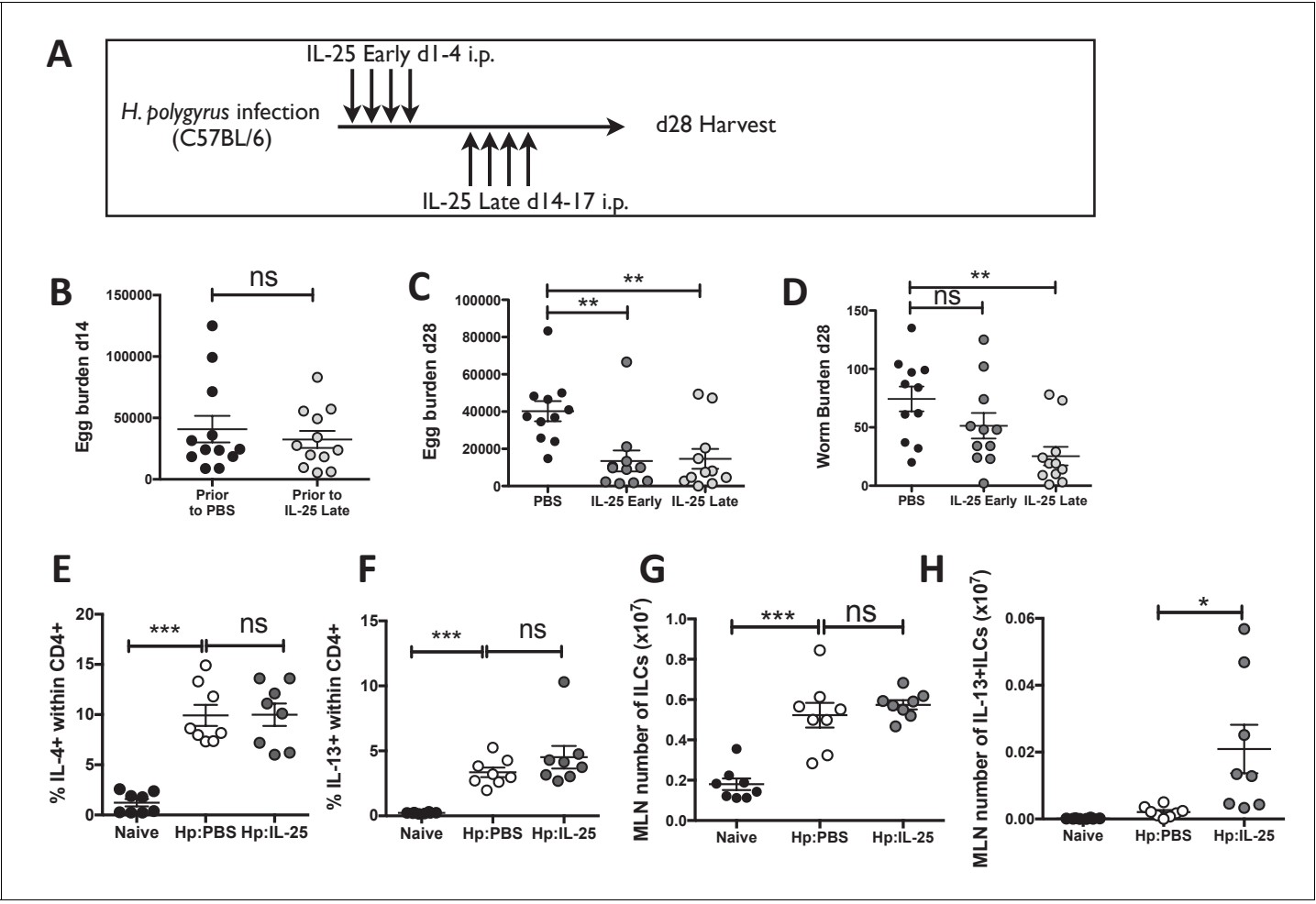

**Figure 4.** IL-25 induces protective immune responses in the late stage of infection. *H. polygyrus*-infected C57BL/6 mice were given 0.4 μg recombinant IL-25 i.p. at day 1–4 (early) or day 14–17 (late) post-infection according to the schedule shown in (A). Before administration of IL-25, intestinal egg burden was analysed in two groups at day 14 post-infection (B). Mice were then analysed at day 28 post-infection for fecal egg counts (C) and intestinal adult worm burden (D) following administration of IL-25 early or late. *H. polygyrus*-infected C57BL/6 mice were given 0.4 μg recombinant IL-25 i.p. at day 14–17 (late) post-infection. Day 18 post-infection, MLN cells underwent ICCS to compare the proportion of IL-4⁺CD4⁺ (E) and IL-13⁺CD4⁺ (F) T cells, as well as the number of ILCs (G) and Lin-IL-13 + ILC2 s (H) by flow cytometry. Results shown are data pooled from three experiments with n ≥ 3 mice/group (B–D), or are representative of two experiments with n = 4 mice/group (E,F) or pooled from two experiments with n = 4 mice/group (G, H). Data were analysed by unpaired *t* test, where *=p≤0.05,**=p≤0.01,***=p≤0.001 and ns = not significant. Error bars represent Standard Error of the Mean.

DOI: https://doi.org/10.7554/eLife.38269.005

infected mice (*Jenkins et al., 2013*). These cells also respond to the alarmins IL-25 and IL-33 to promote type-2 cytokine production and alternative activation, where adoptive transfer of IL-33-activated macrophages has been reported to induce worm expulsion in mice with chronic *H. polygyrus* infection (*Yang et al., 2013*). Eosinophils also respond to IL-25 and are thought to play a role in limiting Th2 responses following *H. polygyrus* infection (*Strandmark et al., 2017*).

Analysis of the peritoneal lavage revealed striking changes within macrophage and eosinophil populations following administration of IL-25 to *H. polygyrus* infected mice. IL-25 injection significantly increased production of IL-13 by Siglec-F$^+$CD11b$^+$F4/80$^+$ monocyte populations following administration of IL-25 to naïve and *H. polygyrus* infected mice (*Figure 5A,B*). Administration of IL-25 also significantly increased expression of the marker of alternative activation RELM-α by naïve macrophages in vivo (in an environment with low IL-4/13 levels), however alternative activation was further increased above and beyond this level in the setting of *H. polygyrus* infection (*Figure 5B,C*). A similar pattern was seen for the number of eosinophils in the PL (*Figure 5D,E*). The number of peritoneal monocytes was also significantly increased in response to IL-25, similar to the setting of *H. polygyrus* infection and a combination of infection and IL-25 (*Figure 5F*). In vitro treatment of bone-marrow-derived macrophages with recombinant cytokines confirmed previous reports that IL-4, but not IL-25, induced expression of the alternatively activated macrophage marker RELM-α (*Rizzo et al., 2012*). However, when combined in vitro, we found a very marked synergy between IL-4 and IL-25 for increased macrophage RELM-α (*Figure 5G,H*), as well as Arginase-1 expression (*Figure 5I,J*); in each case far above that observed for either cytokine alone.

## IL-25R co-operates with IL-4Rα signaling to prime innate immunity for effective clearance of *H. polygyrus*

It has long been known that IL-4Rα-mediated signaling is the pivotal component of the protective immune response to helminth infection, being required to generate the appropriate innate and adaptive type two cellular responses. To test whether IL-25-mediated promotion of immunity to primary *H. polygyrus* infection was entirely dependent upon IL-4Rα, or could mediate an IL-4Rα-independent mode of protection, we administered exogenous IL-25 from days 14–17 of infection to *Il4ra*-deficient mice (*Figure 6A*). Although late IL-25 injection was previously demonstrated to induce adult worm expulsion in C57BL/6 mice (*Figure 4D*), the worm burden of *Il4ra$^{-/-}$* mice was unaffected by administration of IL-25 at this time-point (*Figure 6B*). In a similar manner to C57BL/6 mice (*Figure 5C and D*) IL-25 injection significantly increased the percentage of eosinophils in the PL of *H. polygyrus* infected BALB/c mice by day 18 (*Figure 6C*). Within the BALB/c strain, 100% of macrophages within the PL expressed RELM-α$^+$ by day 18 in *H. polygyrus* infected mice following administration of IL-25 or PBS control, compared to an average of 28% in naïve mice (*Figure 6D*). However, these responses were completely lacking in infected *Il4ra$^{-/-}$* mice treated with IL-25 (*Figure 6C,D*), confirming that the IL-25 induced response is itself wholly dependent on IL-4Rα signaling.

## Discussion

Generation of the type 2 immune response, and protection from helminth parasite infection, requires sustained interaction and participation of both innate and adaptive immune cells (*Allen and Maizels, 2011*; *Grencis, 2015*; *Harris and Loke, 2017*; *Van Dyken et al., 2016*). Alarmins such as IL-25 are potent activators of type 2 immunity, but as we report here, can play an even more important role in stimulating and mobilising effector mechanisms to expel infection. Specifically, we show that adaptive type 2 responses develop normally in the absence of IL-25R but that eosinophil and macrophage responses are impaired, which is likely to explain compromised helminth expulsion in IL25R-deficient mice.

Notably, by day 28 of *H. polygyrus* infection, the *Il17rb$^{-/-}$* genotype was unable to reduce adult worm or egg burdens, despite the expression of comparable adaptive Th2 response features such as IL-4 and IL-13 expression in CD4$^+$ T cells. A requirement for IL-25/IL-25R signaling to expel *H. polygyrus* has previously been reported in other settings in which Th2 response levels appear unaffected, including both a vaccination model (*Hewitson et al., 2015*), and following drug-abbreviated primary infection (*Pei et al., 2016*). Similarly, it has been reported that the resolution of acute *N. brasiliensis* infection is delayed in *Il25$^{-/-}$* mice despite sufficient Th2 cell differentiation (*Mearns et al., 2014*). These results therefore suggest that Th2 differentiation can occur

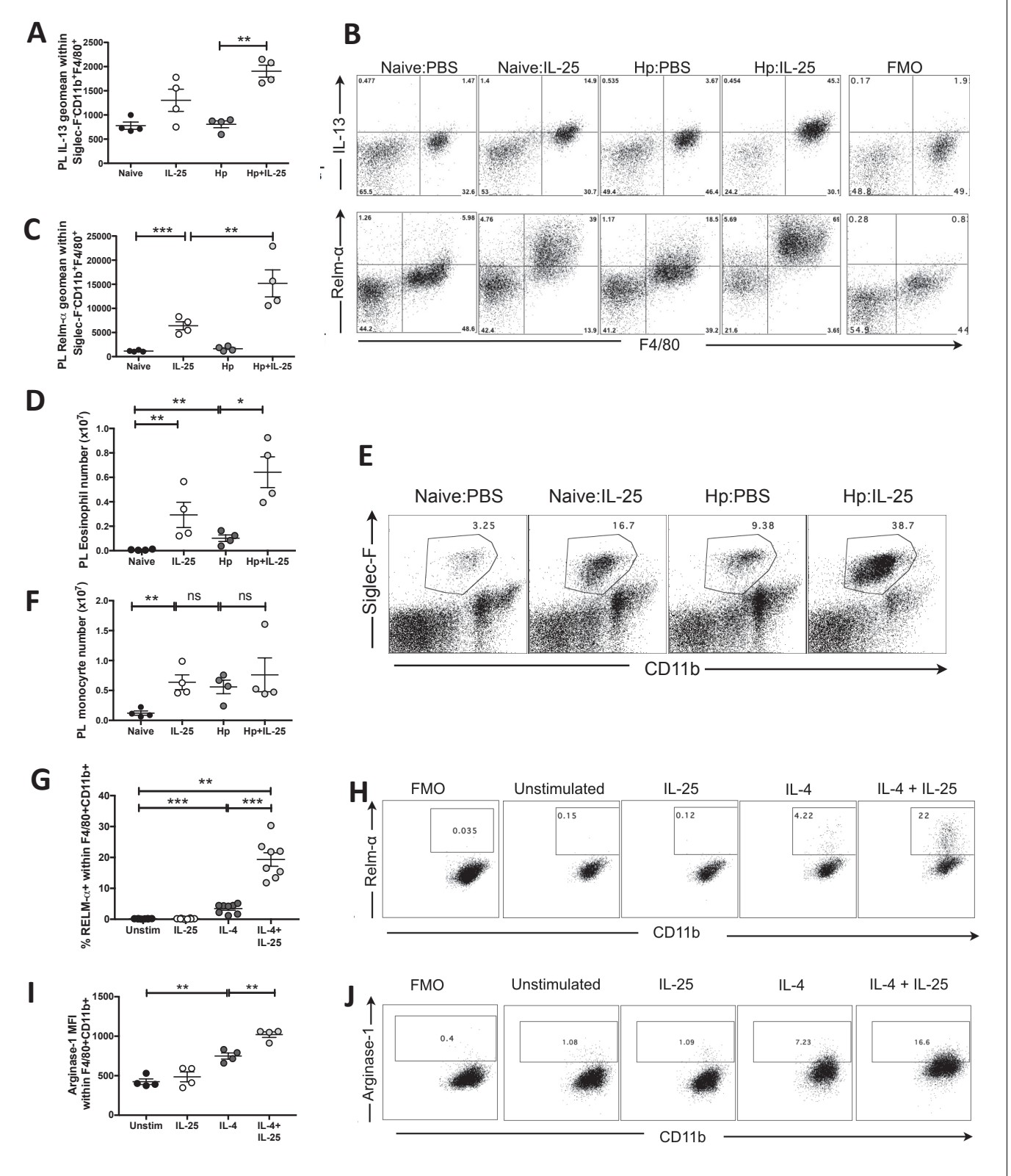

**Figure 5.** IL-25 drives alternative activation and IL-13 expression of macrophages. *H. polygyrus*-infected C57BL/6 mice were given 0.4 μg recombinant IL-25 i.p. at day 14–17 (late) post-infection. At day 18 post-infection, PL cells were taken and stimulated with 10 μg Brefeldin A to determine the expression intensity (geomean) of IL-13 (**A**) or were unstimulated to determine the expression intensity (geomean) of RELM-α (**B, C**) within Siglec-F⁻CD11b⁺F4/80⁺ monocytes by flow cytometry, as shown by the example flow cytometry plots in B and compared to fluorescence minus one (FMO)

*Figure 5 continued on next page*

Figure 5 continued

samples stained from the Hp:IL-25 group. Total Siglec-F$^+$CD11b$^+$ eosinophil (D) numbers were calculated from populations shown in the example flow cytometry plots (E), numbers of Siglec-F$^-$CD11b$^+$ (F) monocytes. Bone marrow-derived macrophages from C57BL/6 mice were generated in vitro and stimulated with 10 ng/ml IL-4, 200 ng/ml IL-25 or a combination of both for 16 hr before analysis of the percentage of RELM-α expression (G, H) and the mean fluorescence intensity (geometric mean) of Arginase-1 expression (I, J) within CD11b$^+$F4/80$^+$ cells by flow cytometry. Results shown are one representative of 2 experiments with n = 4 mice/group (A–F, I, J) or pooled from two experiments with n = 4 replicates/group (G, H). Data were analysed by unpaired t test, where *=p≤0.05,**=p≤0.01,***=p≤0.001 and ns = not significant. Error bars represent Standard Error of the Mean.
DOI: https://doi.org/10.7554/eLife.38269.006

independently of IL-25R signaling following helminth infection and reinforce the idea that innate type 2 responses can operate autonomously from adaptive type 2 responses to control parasite expulsion (*Smith et al., 2012*).

A wide range of cell types respond to IL-25, but we demonstrated by chimera experiments that parasite expulsion requires receptor expression on haematopoietic, rather than non-haematopoietic cells in vivo. As adaptive Th2 responses were comparable in IL-25-deficient and sufficient infected mice, we examined innate populations. In *N. brasiliensis* infections, intestinal tuft cell-derived IL-25 drives parasite expulsion and the activation of ILC2s, in a positive feedback loop mediated by IL-13 (*Gerbe et al., 2016*; *von Moltke et al., 2016*). However, our data indicate that innate lymphoid cells may not be required for immunity to *H. polygyrus*, because anti-CD90 (Thy1) antibody depletion of these cells from *Rag1*-deficient mice did not affect the ability of IL-25 (in combination with IL-4Rα signaling) to expel *H. polygyrus* parasites. Through the use of ST2 staining we were able to demonstrate that both the resident Lin$^-$ST2$^+$CD90$^{hi}$, and the inflammatory Lin$^-$ST2$^-$CD90$^{low}$ ILC populations recently described (*Huang et al., 2015*) are depleted following administration of anti-CD90.2 antibody to *H. polygyrus*-infected mice given IL-25. It is possible that our regime of anti-CD90.2 antibody treatment spares a residual and essential ILC population, but genetic strategies to fully ablate ILCs by deletion of the common gamma chain receptor (γc) are confounded in our system because γc also forms part of the IL-4R.

It has previously been noted that ILC2 numbers show only modest expansion in the MLN of C57BL/6 mice infected with *H. polygyrus* and these can be considerably boosted following administration of IL-25 (*Hepworth et al., 2012*). In confirming this, our data also show an increase in total ILCs with *H. polygyrus* infection, of which only a small proportion are ILC2s. One explanation is that chronic *H. polygyrus* infection limits IL-25 and ILC2 expansion by eliciting the production of IL-1β (*Zaiss et al., 2013*). We have also demonstrated that *H. polygyrus* infection results in the preferential trafficking of LTi-like ILC3s to the MLN (*Mackley et al., 2015*), pointing again to a diminished role of ILC2s in immunity to this infection, while a recent study reported only modest reduction in adult *H. polygyrus* worm burdens in mice in whom ILC2 numbers were expanded fivefold with IL-2:anti-IL-2 complex (*Pelly et al., 2017*).

In contrast, we uncover striking IL-25R-dependent modifications to macrophage and eosinophil populations following *H. polygyrus* infection, in two important respects. Firstly, the expansion of eosinophil and alternatively activated macrophage numbers seen following helminth infection is strongly amplified by IL-25. Secondly, IL-25 was required for the expression of IL-13 within macrophages, in accordance with earlier publications (*Fort et al., 2001*; *Yang et al., 2013*). Macrophages may thus represent a major source of IL-13 in infected animals, suggesting an element of autologous stimulation through the IL-13-responsive IL-4Rα, which is required for macrophage alternative activation.

The alternative activation of macrophages is critical for immunity to *H. polygyrus* (*Anthony et al., 2006*), and we now show that IL-25, in addition to IL-4Rα signaling, is required both for optimal macrophage activation, and for parasite elimination. Thus, for example, the two cytokines synergise dramatically to induce high levels of RELM-α. Consistent with established findings, macrophage activation by IL-25 cannot occur in the absence of IL-4Rα signaling (*Kang et al., 2012*). Interestingly, the production of IL-25 following nematode infection is itself dependent on IL-4Rα, mediated by IL-13 activation of STAT6 (*Zhao et al., 2010*). Furthermore, upregulation of the IL-25 receptor *Il17rb* in the small intestine is STAT-6 dependent (*Zhao et al., 2010*). We are currently investigating whether macrophage expression of the IL-25 receptor is specifically required for parasite expulsion in the presence of IL-4Rα signaling.

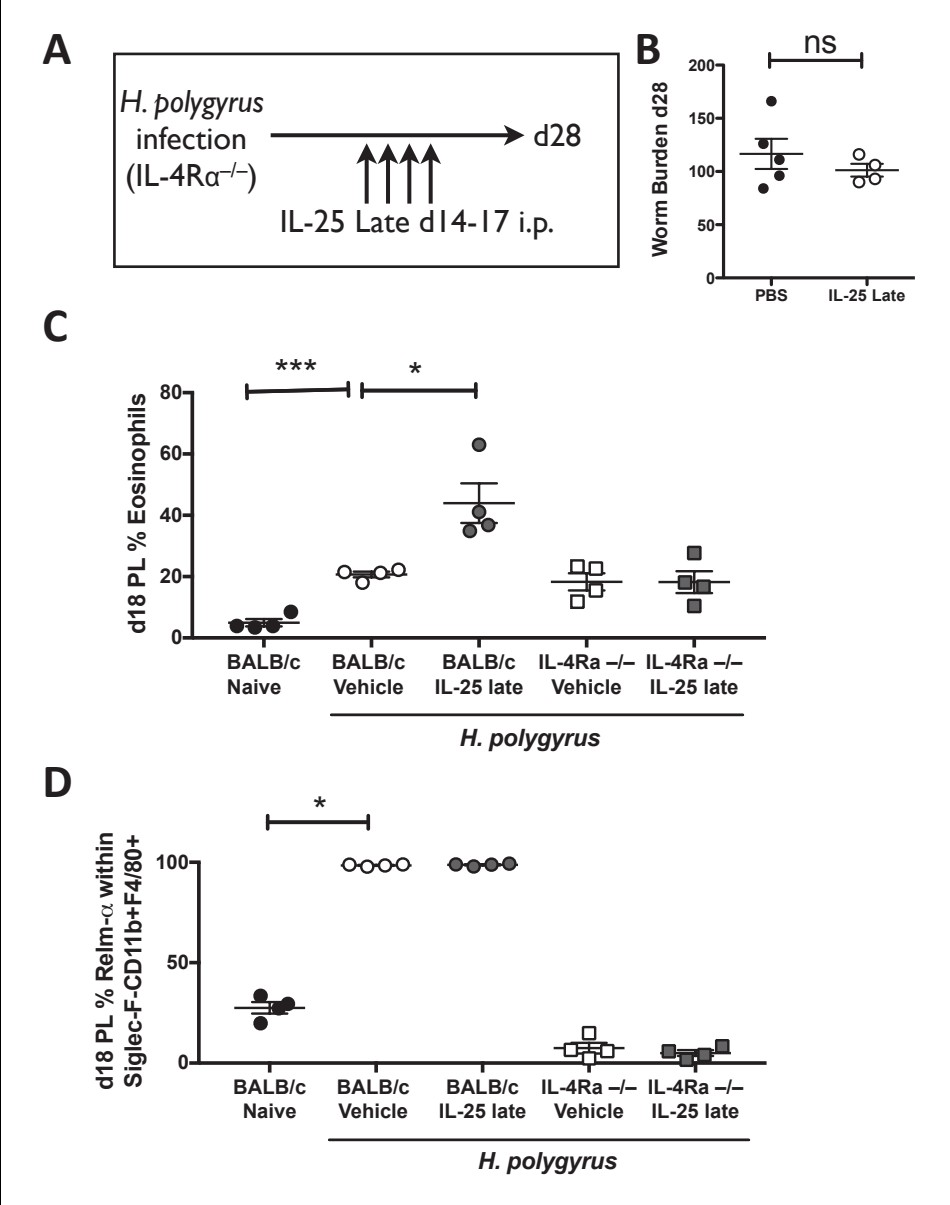

**Figure 6.** IL-4Rα expression is required for activation of the immune system by IL-25. *H. polygyrus*-infected *Il4ra*[-/-] mice (IL-4R[-/-]) were given 0.4 μg recombinant IL-25 i.p. days 14–17 (late) post-infection (**A**). Mice were analyzed at day 28 post-infection for intestinal adult worm burden (**B**). *H. polygyrus*-infected BALB/c and *Il4ra*[-/-] mice (IL-4R[-/-]) were given 0.4 μg recombinant IL-25 i.p. days 14–17 (late) post-infection. Day 18 post-infection, PL cells were taken to determine the percentage of eosinophils (**C**) and percentage of RELM-α expression within Siglec-F[-]CD11b[+]F4/80[+] macrophages by flow cytometry (**D**). Results shown are one representative of 2 experiments with n ≥ 4 mice/group (**A–D**). Data were analysed by unpaired *t* test, where *=p≤0.05,**=p≤0.01,***=p≤0.001 and ns = not significant. Error bars represent Standard Error of the Mean.
DOI: https://doi.org/10.7554/eLife.38269.007

In summary, we demonstrate that the IL-25R is not required for generation of a sufficient Th2 response to helminth infection and that instead, it is required for late effector responses to fully resolve chronic infection. We identify the macrophage and the eosinophil as two prominent populations activated following IL-25 administration, with induction of IL-13 expression within macrophages that is likely, in turn, to drive the suite of intestinal epithelial mechanisms that lead to nematode expulsion (*Cliffe et al., 2005*; *Maizels et al., 2012*; *Patel et al., 2009*). We also clearly demonstrate

an enhancement in macrophage alternative activation, which plays a critical role in nematode immobilisation and killing (*Hewitson et al., 2015*) and suggest that ILC2s may be redundant in driving IL-25R-dependent immunity to chronic helminth infection. These results now place IL-25 in the central role of mobilising innate effector cells other than ILCs, in the context of an IL-4/13-replete environment, to protect against chronic gastrointestinal helminth infection. Future work will elucidate in greater detail the range of responsive cells and their inter-relationship in the network that orchestrates parasite expulsion and immunity.

# Materials and methods

## Key resources table

| Reagent type (species) or resource | Designation | Source or reference | Identifiers |
|---|---|---|---|
| Mus musculus, BALB/c | *Il17rb$^{-/-}$* (IL-25R$^{-/-}$) | *Neill et al. (2010)* Nuocytes represent a new innate effector leukocyte that mediates type-2 immunity. Nature 464:1367–1370 | |
| Mus musculus, C57BL/6 | *Rag1$^{-/-}$* | Mombaerts,P., Iacomini,J., Johnson, R.S., Herrup, K., Tonegawa,S., and Papaioannou, V. E. 1992. Cell. Vol 68: 869–877. PMID: 1547488 | |
| Mus musculus, C57BL/6 | *Il4ra-/-* | Noben-Trauth N, Shultz LD, Brombacher F, Urban JF, Jr., Gu H, Paul WE. 1997. An interleukin 4 (IL-4)-independent pathway for CD4 + T cell IL-4 production is revealed in IL-4 receptor-deficient mice. Proc Natl Acad Sci USA 94:10838–10843. | |
| Parasite | *Heligmosomoides polygyrus bakeri* | *Johnston et al., 2015*. Cultivation of Heligmosomoides polygyrus: an immunomodulatory nematode parasite and its secreted products. Journal of Visualized Experiments 98:e52412. | |
| Antibody | Anti-IL-4 | BioXCell | Clone 11B11 |
| Antibody | Anti-IL-13 | BioLegend | JES10-5A2 |
| Antibody | Anti-CD90.2 | BioXCell | 30H12 |
| Antibody | Anti-SiglecF | BD Pharmingen | E50-2440 |
| Antibody | Anti-CD11b | BioLegend | M1/70 |
| Antibody | Anti-F4/80 | BioLegend | BM8 |
| Antibody | Anti-Ly6G | BioLegend | 1A8 |
| Antibody | Anti-Ly6C | BioLegend | AL-21 |
| Antibody | Anti-RELMα | R and D Systems | 22603 |
| Antibody | Anti-Arginase-1 (Polyclonal) | R and D Systems | IC5868P |
| Antibody | Anti-Lineage | BioLegend | 17A2,RB6-8C5,RA3-6B2,Ter-119,M1/70 |
| Peptide, recombinant protein | IL-4 | Peprotech | 214–14 |
| Peptide, recombinant protein | IL-25 | BioLegend | 587302 |
| Commercial assay or kit | Foxp3 Staining Kit | eBioScience | 88–8118 |

*Continued on next page*

*Continued*

| Reagent type (species) or resource | Designation | Source or reference | Identifiers |
|---|---|---|---|
| Other | Control IgG2b immunoglobulin | BioXcell | LTF-2 |
| Other | Anti-CD90.2 Microbeads | Miltenyi | 130-049-101 |

## Mice

BALB/c, IL-25R$^{-/-}$ (*Il17rb$^{-/-}$*) (*Neill et al., 2010*), C57BL/6, *Rag1$^{-/-}$* mice and *Il4ra$^{-/-}$* mice were bred at the University of Edinburgh. All animal protocols adhered to the guidelines of the UK home office, complied with the Animals (Scientific Procedures) Act 1986, were approved by the Ethical Review Committees of the University of Edinburgh and the University of Glasgow, and were performed under the authority of the UK Home Office Project numbers 60/4105 and 70/8384.

## Construction of bone marrow chimeras

BALB/c or *Il17rb$^{-/-}$* mice were exposed to 11.5 Gy γ radiation administered in two doses before intravenous reconstitution with 2 million bone marrow cells from BALB/c or *Il17rb$^{-/-}$* mice, which had been depleted of CD90$^+$ cells using CD90.2 microbeads (Miltenyi). Eight weeks post-transfer, recipients were infected with 200 *H. polygyrus* by gavage.

## Parasites and antigens

*H. polygyrus bakeri* was maintained and adult *H. polygyrus* E/S (HES) was prepared as described elsewhere (*Grainger et al., 2010*; *Johnston et al., 2015*). Egg counts from individual mice were assessed by weighing the feces before dissolving in 2 ml saturated sodium chloride solution; egg counts were performed using a McMaster chamber and the average number of eggs/g feces calculated per sample. Mice were infected with 200 L3 stage *H. polygyrus* by gavage.

## Preparation and administration of IL-4/anti-IL-4 complexes

A pre-prepared complex of 5 μg rIL-4 (Peprotech) and 25 μg anti-IL-4 (clone 11B11; BioXcell, NH) was administered to mice i.p. (*Urban et al., 1995*).

## Depletion of ILCs

*Rag1$^{-/-}$* mice received 200 μg anti-CD90.2/Thy1.2 (clone 30H12; BioXcell) or a rat IgG2b control (clone LTF-2; BioXcell) i.p. on days 12, 15, 18 and 21 post-infection.

## In vitro Ag-specific restimulation

A single cell suspension was made of MLN before plating cells at $5 \times 10^5$/well in the presence of 2 μg/ml HES and media alone for 72 hr at 37°C/5% CO$_2$. Supernatants were then harvested and analysed for IL-4, IL-5, IL-13 by commercially available ELISA (BD Pharmingen).

## Generation of bone marrow-derived macrophages

Bone marrow was extracted from tibia and femurs of C57BL/6 mice. A single cell suspension was formed in 10 ml of PBS by passing through a 23 g needle, then filtered through a 100 μm nylon cell strainer. Cells were plated at a density of $6 \times 10^6$ cells/ plate on 90 cm Petri dishes in 10 ml cDMEM with 20% L929 media as a source of M-CSF and incubated at 37°C incubator with 5% CO$_2$. A further 5 ml of cDMEM with 20% L929 was added on day 3. Differentiated macrophages were harvested on day 7 using 3 mM EDTA/10 mM glucose in PBS. Cells were washed in PBS, resuspended in cDMEM, plated at $2 \times 10^5$ cells/ well in 96 well plates and stimulated with 10 ng/ml IL-4, 200 ng/ml IL-25 or a combination of both for 16 hr before analysis of the cells by flow cytometry.

## Flow cytometry

All flow cytometry was performed using Becton-Dickinson Canto or LSR-II flow cytometers. For innate cell surface phenotyping, PL or MLN were stained with a combination of antibodies to

Siglec-F (E50-2440), CD11b (M1/70), F4/80 (BM8), Ly6G (1A8) and Ly6C (AL-21 or HK1.4). Following fixation and permeabilisation with the Foxp3 staining kit (eBioscience) cells underwent intracellular staining with RELM-$\alpha$ (RnD Systems) followed by zenon anti-rabbit A647 (Invitrogen) and FITC-conjugated anti-human Ki67 (BD Biosciences). For intracellular cytokine staining of monocytes, $0.5-1 \times 10^6$ PL cells were incubated with 10 µg/ml Brefeldin A for 4 hr. Following cell surface staining as above, and fixation and permeabilization with Fix/Perm buffer (BD Pharmingen), cells underwent intracellular staining with anti-IL-13 (JES10-5A2). For intracellular staining of lymphocytes, MLNCs were incubated with 0.5 µg/ml PMA and 1 µg/ml ionomycin for 1 hr before the addition of 10 µg/ml Brefeldin A for a further 3 hr. Staining was performed by re-suspending cells in a combination of Abs to CD4 (GK1.5), ICOS (DX29), and the following combination to define Lin⁻: CD3 (17A2), CD5 (53–7.3), CD8$\alpha$ (RPA-T8), CD49b (DX5), CD11c (HL3), F4/80 (BM8), CD19 (eBio1D3), Gr-1 (RB6-8C5), TCR$\beta$ (H57-597) and CD11b (M1/70).

## Statistical analysis

Data were assessed for normality and equal variance and were log transformed if required; all data passed these criteria and an unpaired $t$ test was used or, where more that three groups were tested, a parametric one-way ANOVA followed by Tukey's multiple comparison test was used.

## Acknowledgements

We thank Judi Allen and Dominik Rückerl (University of Manchester, UK) for $Rag1^{-/-}$ and $Il4ra^{-/-}$ mice. We thank the Wellcome Trust for support through a Senior Investigator Award to RMM for SL (Ref 106122), the European Commission for support through a Marie Skłodowska-Curie global fellowship to KAS, an Edinburgh Clinical Academic Track (ECAT) Studentship for FV, and for core support from the Wellcome Trust through the Wellcome Centre for Molecular Parasitology (Ref 104111).

## Additional information

### Funding

| Funder | Grant reference number | Author |
| --- | --- | --- |
| Wellcome | 106122 | Rick M Maizels |
| Wellcome | 090281 | Rick M Maizels |
| European Commission | 657639 | Katherine A Smith |

The funders had no role in study design, data collection and interpretation, or the decision to submit the work for publication.

### Author contributions

Katherine A Smith, Conceptualization, Formal analysis, Investigation, Methodology, Writing—original draft, Writing—review and editing; Stephan Löser, Henry J McSorley, Investigation, Writing—review and editing; Fumi Varyani, Yvonne Harcus, Investigation; Andrew NJ McKenzie, Resources, Writing—review and editing; Rick M Maizels, Conceptualization, Supervision, Funding acquisition, Writing—original draft, Project administration, Writing—review and editing

### Author ORCIDs

Katherine A Smith (iD) https://orcid.org/0000-0001-8150-5702
Rick M Maizels (iD) http://orcid.org/0000-0003-3216-1944

### Ethics

Animal experimentation: All animal protocols adhered to the guidelines of the UK home office and complied with the Animals (Scientific Procedures) Act 1986. The protocols were approved by the Ethical Review Committees of the University of Edinburgh (UK Home Office Project number 60/4105) and the University of Glasgow (Project number 70/8384).

**Decision letter and Author response**
Decision letter https://doi.org/10.7554/eLife.38269.010
Author response https://doi.org/10.7554/eLife.38269.011

## Additional files

### Supplementary files
• Transparent reporting form
DOI: https://doi.org/10.7554/eLife.38269.008

### Data availability
All data generated or analysed during this study are included in the manuscript and supporting files.

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
