## [Decision Letter]

Thank you for submitting your article "Concerted IL-25R and IL-4Rα signaling drive innate type 2 effector immunity and helminth expulsion independently of ILCs" for consideration by *eLife*. Your article has been reviewed by three peer reviewers, including Andrew J MacPherson as the Reviewing Editor and Reviewer #1, and the evaluation has been overseen by Tadatsugu Taniguchi as the Senior Editor.

The reviewers have discussed the reviews with one another and the Reviewing Editor has drafted this decision to help you prepare a revised submission.

The paper shows that IL-25 stimulates late clearance of the helminth *Heligmosomoides polygyrus* in synergy with IL-4. These results have been obtained in a series of timed cytokine/cytokine complex supplementation experiments, including in the context of IL-4R deficient strain combinations, using radiation chimeras to show that IL-25R is required on hematogenous cells. A depletion approach was used to show that innate lymphoid cells in RAG mice are not required for the phenotype.

The work is discussed in the context of different helminths and the context of different IL-25-responsive cell types. Data in the paper shows activation of (alternatively activated) macrophages and eosinophils with IL-13 expression as an effector cytokine.

The reviewers were generally positive about the paper and its conclusions. The data regarding the role of IL-4 and IL-25 were considered interesting although both cytokines have been shown before to play a role in parasite resistance. The novelty was considered to be that the immune effector cell is an innate cell and that it does not belong to the group of innate lymphoid cells (ILC). The key experiment to determine this is the case was using an antibody to CD90 to remove innate lymphoid cells in vivo. This approach, however, was considered generally not to be a definitive approach to defining whether ILC are involved or not. The data presented does show that there are changes in the spleen and peritoneum after anti-CD90 antibody treatment of infected mice but this does not mean that this is effective at the sites related to intestinal infection i.e. the gut tissue itself or mesenteric lymph node. Moreover, the authors refer to a paper by Huang et at, 2015 stating that inflammatory ILC2 (and the likely ILC involved) express high levels of CD90 and therefore would be removed by such treatment. In fact the ILC2 were shown to express low levels of CD90 as compared to natural ILC2 and as such are unlikely to be efficiently cleared by anti-CD90 treatment. Thus the current conclusions made that it is a non ILC population of innate cells that are responsible for the protection observed may be premature. This could be tested in either background strain comparing responses in RAG null mice with RAG cγ chain null mice together with appropriate treatments. In fact, Figure 4H is supportive of IL-25 up-regulating ILC2 and this correlates with protection so these experiments will be key to addressing the main conclusions of the paper.

Essential revisions:

The key point would be to convincingly show that the cells that mediate the changes are not an ILC population – since this is a critical part of the paper and its conclusions. Ideally this should be by experiments in vivo together with better phenotyping of the population. An alternative would be to change the title and the major conclusion of the paper – and accept that the effects seen could be ILC2 mediated. This would however reduce the conceptual advance that you are proposing.

[Editors' note: further revisions were requested prior to acceptance, as described below.]

Thank you for sending your revised article entitled "Concerted IL-25R and IL-4Rα signaling drive innate type 2 effector immunity for optimal helminth expulsion independently of innate lymphoid cells." for peer review at *eLife*. Your article has been re-evaluated by a Reviewing Editor, and the evaluation is being overseen by Tadatsugu Taniguchi as the Senior Editor.

In our original decision requesting a revision we stated that the key point would be to convincingly show that the cells that mediate the changes are not an ILC population – since this is a critical part of the paper and its conclusions. Ideally this should be by experiments in vivo together with better phenotyping of the population. An alternative would be to change the title and the major conclusion of the paper – and accept that the effects seen could be ILC2 mediated. This would however reduce the conceptual advance that you are proposing.

In the evaluation of the revision we have the following concerns. Other publications have been leveraged to substantiate the claim that anti-CD90.2 treatment will clear all ILC2. The experimental protocols in the cited papers are not the same as were used here and the steady state data may not apply here to the critical groups. Also the data in Figure 3 shows that whilst antibody treatment does significantly reduce levels of ILC there are still ILC remaining. Data is presented from the peritoneum and spleen not the site of parasite infection which is the intestine. It is stated that you did not look in the intestinal tissue as it was needed it for parasite enumeration. This is understood although it is felt that the experiment still needs to be undertaken to support your hypothesis together with experiments in RAGcgamma chain deficient mice to prove ILC have no role. Whilst the results from these extra experiments may well show that the current conclusions are correct i.e. independent of ILC, but the current data was not thought definitive on this point, and without the data from the extra experiments, the present data was considered suggestive but not sufficiently conclusive to justify the current strength of the title and the interpretations.

---

## [Author Response]

[…] The reviewers were generally positive about the paper and its conclusions. The data regarding the role of IL-4 and IL-25 were considered interesting although both cytokines have been shown before to play a role in parasite resistance. The novelty was considered to be that the immune effector cell is an innate cell and that it does not belong to the group of innate lymphoid cells (ILC). The key experiment to determine this is the case was using an antibody to CD90 to remove innate lymphoid cells in vivo. This approach, however, was considered generally not to be a definitive approach to defining whether ILC are involved or not. The data presented does show that there are changes in the spleen and peritoneum after anti-CD90 antibody treatment of infected mice but this does not mean that this is effective at the sites related to intestinal infection i.e. the gut tissue itself or mesenteric lymph node. Moreover, the authors refer to a paper by Huang et at, 2015 stating that inflammatory ILC2 (and the likely ILC involved) express high levels of CD90 and therefore would be removed by such treatment. In fact the ILC2 were shown to express low levels of CD90 as compared to natural ILC2 and as such are unlikely to be efficiently cleared by anti-CD90 treatment. Thus the current conclusions made that it is a non ILC population of innate cells that are responsible for the protection observed may be premature. This could be tested in either background strain comparing responses in RAG null mice with RAG cγ chain null mice together with appropriate treatments. In fact, Figure 4H is supportive of IL-25 up-regulating ILC2 and this correlates with protection so these experiments will be key to addressing the main conclusions of the paper.

Regarding the first issue, of whether gut tissue ILCs are equally depleted, we would refer to published studies in which anti-CD90 depletion removes ILCs within both the intestine and mLN (Sonnenberg et al., 2012 Science 336:1321-1325), and to emphasise that our protocol delivering anti-CD90.2 at days 12, 15, 18 and 21 post-infection should mediate efficient removal of all resident ILCs within the lymphoid and intestinal tissue, particularly as all resident ILCs are reported to express high levels of CD90 at steady state (Kim et al., 2015 Immunity 43:107; Hepworth et al., 2013 Nature 498:113-117).

We could not directly evaluate intestinal tissue populations as this tissue was used to evaluate parasite burdens. In addition, the lymph nodes of RAG mice used in these experiments are small and lymphopenic (Seymour et al., 2006 Vet Path 43:401-423) and could not be reliably analysed by flow cytometry.

On the second question of iILC2 expression of CD90/Thy1, we accept that our wording in the Discussion of the manuscript was imprecise; Huang et al., 2015, describes two populations of ILC that are induced in response to alarmin administration, IL-25 responsive Lin^-^IL-7Ra^+^Thy1^low^ST2-KLRG1^hi^ iILC2s; and resident and IL-33 responsive Lin^-^IL-7Ra^+^Thy1^hi^ST2+KLRG1^int^ nILC2s. We now modify our text in the third paragraph of the Discussion in order to clarify this point.

In our gating strategy for Figure 3, cells from RAGKO mice were stained for lineage (CD3, CD5, CD8a, CD11c, CD19, DX5, F4/80, GR-1), TCRβ and CD11b, CD45, ICOS, ST2 and CD4, allowing us to gate on all lineage negative populations and distinguish ST2^+^ (nILC2) from ST2^-^ (iILC) subsets. Using this gating strategy, we find that all populations are depleted following administration of anti-CD90.2 in both the spleen and peritoneal lavage. We now include these data in Figure 3 (Panels E, F, G) and state this in the last paragraph of the subsection “Effective clearance of adult worms in immune-deficient mice requires IL-25R and IL-4Rα signaling through the innate immune compartment, but acts independently of ILCs” and in the third paragraph of the Discussion.

Essential revisions:The key point would be to convincingly show that the cells that mediate the changes are not an ILC population – since this is a critical part of the paper and its conclusions. Ideally this should be by experiments in vivo together with better phenotyping of the population. An alternative would be to change the title and the major conclusion of the paper – and accept that the effects seen could be ILC2 mediated. This would however reduce the conceptual advance that you are proposing.

We address this concern above, by clarifying our statement referring to Huang et al., 2015 in the third paragraph of the Discussion, and including the data demonstrating that the antibody depletion approach efficiently removed all ILC subsets from our mice. As these mice remained able to expel parasites, we feel that the title and conclusion of the paper remain valid.

[Editors' note: further revisions were requested prior to acceptance, as described below.]

[…] In the evaluation of the revision we have the following concerns. Other publications have been leveraged to substantiate the claim that anti-CD90.2 treatment will clear all ILC2. The experimental protocols in the cited papers are not the same as were used here and the steady state data may not apply here to the critical groups. Also the data in Figure 3 shows that whilst antibody treatment does significantly reduce levels of ILC there are still ILC remaining. Data is presented from the peritoneum and spleen not the site of parasite infection which is the intestine. It is stated that you did not look in the intestinal tissue as it was needed it for parasite enumeration. This is understood although it is felt that the experiment still needs to be undertaken to support your hypothesis together with experiments in RAGcgamma chain deficient mice to prove ILC have no role. Whilst the results from these extra experiments may well show that the current conclusions are correct i.e. independent of ILC, but the current data was not thought definitive on this point, and without the data from the extra experiments, the present data was considered suggestive but not sufficiently conclusive to justify the current strength of the title and the interpretations.

We are again grateful to the editors and reviewers for their constructive criticism and final positive evaluation. We have amended the text as follows to remove any categorical statements about the role of ILC2s, in concordance with the above advice:

· Title: removed "independently of innate lymphoid cells"

· Abstract: amended the final two sentences to avoid claiming ILCs are totally absent

· Results: deleted *"*but acts independently of ILCs"

· Results: inserted reference to Pelly et al. who showed ILC2 transfer was ineffective

· Discussion: amended the sentence stating "we were able to exclude innate lymphoid cells" to a more reserved "our data indicate that ILCs may not be required"

· Discussion: inserted new sentence acknowledging the shortcoming of anti-CD90 antibody treatment and discussing the problem with γc deficient mice in our system

· Discussion: added reference to the Pelly et al. paper

· Discussion: replaced "establish that ILC2s are redundant" with "suggest that ILC2s may be redundant"

· Figure 3 Legend: removed "independently of ILCs"